# Predictors of an early death in patients diagnosed with colon cancer: a retrospective case–control study in the UK

Conan Donnelly,[1] Nigel Hart,[2] Alan David McCrorie,[3] Michael Donnelly,[4] Lesley Anderson,[4] Lisa Ranaghan,[5] Anna Gavin[6]

[1]University of Cork, National Cancer Registry Ireland, Cork, Ireland
[2]School of Medicine, Dentistry and Biomedical Sciences, Queen's University Belfast, Belfast, UK
[3]Centre for Cancer Research and Cell Biology, Queen's University Belfast, Belfast, UK
[4]Centre for Public Health, Queen's University Belfast, Belfast, UK
[5]Belfast City Hospital, Belfast Health and Social Care Trust, Belfast, UK
[6]N Ireland Cancer Registry, Queen's University Belfast, Belfast, UK

**Correspondence to**
Dr Conan Donnelly;
C.donnelly@nicr.ie

## ABSTRACT

**Objective** Despite considerable improvements, 5-year survival rates for colon cancer in the UK remain poor when compared with other socioeconomically similar countries. Variation in 5-year survival can be partly explained by higher rates of death within 3 months of diagnosis in the UK. This study investigated the characteristics of patients who died within 3 months of a diagnosis of colon cancer with the aim of identifying specific patient factors that can be addressed or accounted for to improve survival outcomes.

**Design** A retrospective case–control study design was applied with matching on age, sex and year diagnosed. Patient, disease, clinical and service characteristics of patients diagnosed with colon cancer in a UK region (2005–2010) who survived less than 3 months from diagnosis (cases) were compared with patients who survived between 6 and 36 months (controls). Patient and clinical data were sourced from general practice notes and hospital databases 1–3 years prediagnosis.

**Results** Being older (aged ≥78 years) and living in deprivation quintile 5 (OR=2.64, 95% CI 1.15 to 6.06), being unmarried and living alone (OR=1.64, 95% CI 1.07 to 2.50), being underweight compared with normal weight or obese (OR=3.99, 95% CI 1.14 to 14.0), and being older and living in a rural as opposed to urban area (OR=1.96, 95% CI 1.21 to 3.17) were all independent predictors of early death from colon cancer. Missing information was also associated with early death, including unknown stage, histological type and marital/accommodation status after accounting for other factors.

**Conclusion** Several factors typically associated with social isolation were a recurring theme in patients who died early from colon cancer. This association is unexplained by clinical or diagnostic pathway characteristics. Socially isolated patients are a key target group to improve outcomes of the worst surviving patients, but further investigation is required to determine if being isolated itself is actually a cause of early death from colon cancer.

## INTRODUCTION

Despite considerable improvements, UK survival rates for colon cancer remain poor by international comparison, with higher

## Strengths and limitations of this study

► The study sample was generated from a high-quality population-based cancer registry system with relatively few death certificate only cases.
► Case–control design provided an efficient method of collecting data and allowed the development of a control group that was matched on important non-modifiable characteristics.
► Data used in this study predate the introduction of the national bowel cancer screening programme in this UK region, which should mitigate any improvement in survival independently associated with bowel cancer screening.
► Survival of controls was restricted to a population of patients whose survival was less than 3 years and similar to the case population.
► The study identified several characteristics which discriminated between cases and controls suggesting that patients who die within the first few months of diagnosis are a specific patient cohort who requires attention.

5-year survival reported in Norway, Sweden, Canada and Australia[1] and poorer survival in the UK compared with several countries reported in Eurocare.[2] These deficits have largely been explained by survival at 3 months postdiagnosis.[3] Patients who survive beyond this period in the UK have similar 5-year survival rates to their counterparts in better performing countries.[1] Approximately 19% of patients with colorectal cancer in the UK and 16% in Northern Ireland (NI) died within 3 months of diagnosis between 2006 and 2008.[4 5] It was estimated that if survival in England matched that of Norway, 13.6% fewer patients would die within the 3-month period.[3] Generally poor survival is linked with a number of factors, including late-stage disease at diagnosis,[6] poor patient fitness due to coexisting disease,[7] and limited

availability of and access to high-quality investigations and treatment.[8]

Reasons for diagnostic delay in colorectal cancer are well documented.[9] Lower educational status[10 11] and rural residence[12 13] have been associated with delayed help-seeking. Additionally, stronger social networks have been associated with shorter diagnostic delay.[14–16] Clinical characteristics also play a role. Patients with comorbid disease[11 17] and/or multiple symptoms[11] are less likely to delay compared with those with non-specific symptoms.[17 18] Application of referral guidelines by general practitioners (GPs) has been shown to reduce delay,[9] while younger patients,[13 19] those of lower socioeconomic status[20] and frequent help-seekers[10 16] were less likely to be referred. While bowel cancer screening was introduced in the UK in 2007[21] and in NI in 2011,[22] the vast majority patients are diagnosed clinically[23]; therefore, the role of clinical decision making in early colon cancer diagnosis remains paramount.

The relationship between these factors and surviving past the first few months following a colon cancer diagnosis has not been adequately investigated, and their role in explaining international survival differences requires attention. The aim of this study was to investigate patient, clinical and disease factors associated with early death in patients with colon cancer in NI and to determine factors which might help to identify subgroups in the population for early diagnosis interventions.

## METHODOLOGY

This study employed a retrospective, individually matched, case–control design involving a posthumous review of primary care physician or GP and electronic secondary care notes. The study design was guided by the principles of the Aarhus statement on early diagnosis research.[24] Principles adhered to in this study include items 1–4, 7–9 and 20 of the Aarhus checklist. Date of initial cancer diagnosis is defined by the Northern Ireland Cancer Registry (NICR) as date of first tissue diagnosis in secondary care, not as symptom presentation in primary and/or secondary care.

### Case and control definition and identification
#### Cases
Patients diagnosed with primary colon cancer (International Classification of Diseases, 10th Revision: C18) in NI between January 2005 and December 2010 (prior to the introduction of the national bowel cancer screening programme in this region) were identified using the NICR. Using death registrations, provided by the General Registrar Office, the status and survival of patients were determined. Cases were defined as patients with an observed survival of under 90 days following diagnosis date (as assigned by the NICR). A random sample of all eligible cases were selected using random number tables based on predefined power calculations.

#### Controls
Controls included patients with an observed survival lasting over 6 months and less than 3 years, leaving a 3-month buffer between the survival rates of cases and controls. Controls were individually matched, using individual nearest neighbour matching,[25 26] to cases by age (within 5-year age bands), sex and year of diagnosis (within 2-year groups). In both groups, patients with incident cancer identified by death certificate only (DCO) and patients with recurrence of a previous incident colon cancer were excluded.

### Exposure variables and covariates
Data items were identified through literature review, with items and categorisation defined in consultation with a clinical adviser, GP, a colorectal surgeon and an oncologist. Items were classified into seven areas: demographic factors acting as a surrogate for social isolation (marital status, accommodation status, NI Multiple Deprivation Measure quintile, rural/urban status), lifestyle (smoking and alcohol status, health-seeking activity including uptake of influenza vaccine and frequency of GP attendance) and comorbidities (Charlson Comorbidity Index (CCI) score and psychiatric illnesses). These characteristics were collected from information recorded between 1 and 3 years before diagnosis. Marital and accommodation status were merged in the final analysis due to multicollinearity.

Disease characteristics included symptoms in the year to diagnosis, and disease stage at diagnosis with histology, morphology and grade collected from pathology records held in the NICR, and GP and hospital episodes (including symptoms (classified as 'vague' or 'alarm' based on the National Institute for Health and Care Excellence Guidelines for Suspected Cancer Referral), clinician actions (number of GP episodes before diagnosis and referral) and investigations ordered). In addition, treatment (first treatment type, treatment intent, surgical resection (yes/no), radiotherapy (yes/no), chemotherapy (yes/no)) and death (date, place and cause of death) information was collected. Data were collected by two trained data abstractors under the guidance of a medically trained clinical adviser using a common bespoke proforma. Data were sourced from GP records, electronic hospital records including the hospital discharge records, multidisciplinary team and oncology data systems. Assuming β=0.8 and a two-sided test with a significance level of 5%, a sample size of n=960 (480 cases matched to 480 controls) sufficiently powers this study to detect an OR of 2.1 for any risk factor with a prevalence of 5%, an OR of 1.8 for any risk factor with a prevalence of 10%, and an OR of 1.6 for any risk factor with a prevalence of 15%.

### Statistical methods
Data were analysed using STATA V.14. All missing data were categorised as unknown and included in the analysis. Univariate analysis involved cross-tabulation of all

categorical variables with case/control status. Conditional logistic regression was used to produce unadjusted ORs and associated 95% CIs to identify independent factors associated with early death. Patient characteristics that were deemed to be clinically significant and/or statistically significant at p<0.25 were included in a minimally adjusted multivariable model to test independence from other comorbidities, patient and disease characteristics. Stage and pathway to diagnosis characteristics (number of accident and emergency (A&E) and GP episodes in the 3 months preceding the diagnosis) were added to the models to assess the degree to which they explained variation in early death among different patient groups. Age (a binary classification around the median age (78 years; IQR=19) of cases) and sex-stratified univariate and multivariable analyses were undertaken to investigate differences in patterns in early death between these groups.

### Patient and public involvement
Members of the public, including patients, were not involved in the design or analysis stages of this piece of non-interventional research, but research question was designed to explore the characteristics of patients who die early after a diagnosis of colon cancer.

### RESULTS
There were 4358 colon cancer tumours between 2005 and 2010 registered by the NICR. Of these, 743 (17%) related to patients who died within 3 months of diagnosis and 1069 related to patients who died between 6 months and 3 years. Following exclusions and sampling (figure 1), 484 cases and the same number of matched controls were generated. There were no significant differences between cases included in the study and those not included (online supplementary table 1) regarding stage at diagnosis, deprivation quintile, age and survival. However, the study group included significantly more men than women, as well as fewer patients diagnosed in 2009 and 2010 due to resource constraints in data collection.

### Univariate analysis
Compared with married patients, the odds of early death were higher among single, widowed and those with unknown marital status (table 1). Those who lived alone, in nursing or residential care, or were living with another relative were more likely to die within 3 months compared with those living with a spouse/partner. The odds of early death were also higher in the most deprived communities (23%) compared with the least deprived (13%) (table 1).

Baseline GP consultation activity was not associated with early death. Regarding influenza vaccine uptake, it was not possible to identify patients who were invited for vaccination, although based on age alone 86% of cases were eligible. Approximately 70% took the vaccine at least once during the period of 1–3 years before diagnosis; 15% did not attend for their vaccination and attendance for the remaining 18% was unknown. Patients who

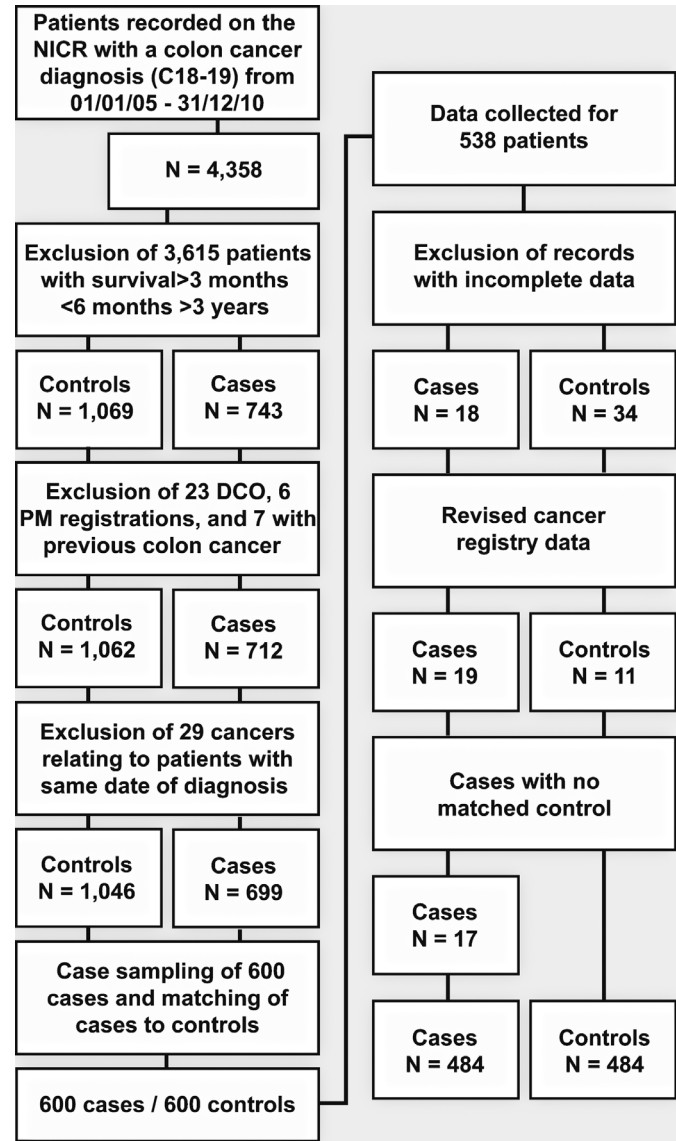

**Figure 1** Case inclusions and exclusions. DCO, death certificate only; NICR, Northern Ireland Cancer Registry; PM, post mortem.

attended twice in the period of 1–3 years before diagnosis had lower odds of early death than patients who did not attend. Smoking status was not significantly associated with early death, with the exception of those who had an unknown smoking status (see table 1).

Being underweight (body mass index (BMI) <18.5) was strongly associated with early death compared with patients with a normal or elevated BMI. However, obesity was not associated with early death when compared with being non-obese (table 2). Comorbidity was common among patients who died early. Almost three-quarters (72%) had at least one comorbidity. CCI score was, however, not associated with early death when means were compared by way of t-test (the mean CCI score for cases was 4.75 compared with 4.90 for controls). Dementia was the only comorbidity within the CCI that was associated with early death—it was present in 8% of cases compared with 4% of controls (table 2).

**Table 1** Demographic characteristics of cases and controls and associated ORs for early death and 95% CIs

| Characteristics | Case n | % | Control n | % | OR* | 95% CI |
|---|---|---|---|---|---|---|
| **Accommodation status** | | | | | | |
| Spouse/Partner | 156 | 32.2 | 233 | 48.1 | 1 | |
| Nursing/Residential | 48 | 9.9 | 23 | 4.8 | 3.93 | 2.18 to 7.09 |
| Sheltered dwelling | 8 | 1.7 | 14 | 2.9 | 0.91 | 0.37 to 2.24 |
| Alone | 156 | 32.2 | 152 | 31.4 | 1.74 | 1.23 to 2.45 |
| Lives with relative | 53 | 11.0 | 40 | 8.3 | 2.32 | 1.41 to 3.82 |
| Unknown | 63 | 13.0 | 22 | 4.6 | 5.32 | 3.00 to 9.43 |
| **Marital status** | | | | | | |
| Married/Cohabiting | 189 | 39.1 | 257 | 53.1 | 1 | |
| Divorced | 15 | 3.1 | 13 | 2.7 | 1.51 | 0.66 to 3.45 |
| Unknown | 44 | 9.1 | 15 | 3.1 | 3.79 | 2.06 to 6.96 |
| Single | 74 | 15.3 | 53 | 11.0 | 1.88 | 1.25 to 2.84 |
| Widowed | 162 | 33.5 | 146 | 30.2 | 1.60 | 1.14 to 2.22 |
| **Urban/Rural status** | | | | | | |
| Rural | 178 | 63.2 | 166 | 65.7 | 1 | |
| Urban | 306 | 36.8 | 318 | 34.3 | 0.89 | 0.68 to 1.17 |
| **Deprivation quintile** | | | | | | |
| Q1 (least deprived) | 64 | 13.2 | 89 | 18.4 | 1 | |
| Q2 | 95 | 19.6 | 92 | 19.0 | 1.45 | 0.76 to 1.74 |
| Q3 | 106 | 21.9 | 111 | 22.9 | 1.01 | 0.66 to 1.54 |
| Q4 | 110 | 22.7 | 102 | 21.1 | 1.49 | 0.99 to 2.24 |
| Q5 (most deprived) | 109 | 22.5 | 90 | 18.6 | 1.47 | 0.95 to 2.27 |
| **Influenza vaccination uptake** | | | | | | |
| No uptake | 71 | 14.7 | 47 | 9.7 | 1 | |
| ≥1 vaccination | 324 | 66.9 | 345 | 71.3 | 0.63 | 0.43 to 0.94 |
| Unknown | 89 | 18.4 | 92 | 19.0 | 0.65 | 0.40 to 1.05 |
| **Baseline consultation activity (tercile)** | | | | | | |
| <11 | 149 | 30.8 | 156 | 32.2 | 1 | |
| 11–19 | 148 | 30.6 | 162 | 33.5 | 0.98 | 0.71 to 1.34 |
| ≥20 | 187 | 38.6 | 166 | 34.3 | 1.20 | 0.87 to 1.67 |
| **Smoking** | | | | | | |
| Non-smoker | 221 | 45.7 | 219 | 15.7 | 1 | |
| Ex-smoker | 141 | 29.1 | 175 | 36.2 | 0.76 | 0.55 to 1.04 |
| Current smoker | 91 | 18.8 | 76 | 15.7 | 1.19 | 0.81 to 1.75 |
| Unknown | 31 | 6.4 | 14 | 2.9 | 2.30 | 1.16 to 4.56 |
| **Alcohol consumption** | | | | | | |
| Current drinker | 154 | 31.8 | 175 | 36.2 | 1 | |
| Ex-drinker | 31 | 6.4 | 26 | 5.4 | 1.37 | 0.78 to 2.43 |
| Never drank | 189 | 39.1 | 191 | 39.5 | 1.13 | 0.82 to 1.55 |
| Unknown | 110 | 22.7 | 92 | 19.0 | 1.38 | 0.96 to 1.99 |

*Conditional logistic regression ORs for cases and controls matched on age, sex and year diagnosed.

Regarding pre-existing psychiatric conditions, 1% of cases were recorded as having schizophrenia, 1% with a learning disability and 13% with anxiety or depression. None were significantly associated with early death. However, the small number of patients with 'other' psychiatric conditions had higher odds of early death

**Table 2** Presence of individual comorbidities included in the Charlson score in cases and controls and associated clogit ORs and 95% CIs

| Characteristics | Case | | Control | | OR* | 95% CI |
|---|---|---|---|---|---|---|
| | n | % | n | % | | |
| **Underweight** | | | | | | |
| No | 467 | 96.5 | 479 | 99.0 | 1 | |
| Yes | 17 | 3.5 | 5 | 1.0 | 3.4 | 1.25 to 9.22 |
| **Obese** | | | | | | |
| No | 425 | 87.8 | 420 | 86.8 | 1 | |
| Yes | 59 | 12.2 | 64 | 13.2 | 0.91 | 0.63 to 1.33 |
| **Dementia** | | | | | | |
| No | 445 | 91.9 | 462 | 95.5 | 1 | |
| Yes | 39 | 8.1 | 22 | 4.6 | 1.85 | 1.07 to 3.19 |
| **Hypertension** | | | | | | |
| No | 279 | 57.6 | 260 | 53.7 | 1 | |
| Yes | 205 | 42.4 | 224 | 46.3 | 0.84 | 0.6 to 1.10 |
| **Ischaemic heart disease** | | | | | | |
| No | 391 | 80.8 | 386 | 79.8 | 1 | |
| Yes | 93 | 19.2 | 98 | 20.3 | 0.94 | 0.68 to 1.29 |
| **Parkinson's disease** | | | | | | |
| No | 473 | 97.7 | 477 | 98.6 | 1 | |
| Yes | 11 | 2.3 | 7 | 1.5 | 1.57 | 0.61 to 4.05 |
| **Valvular heart disease** | | | | | | |
| No | 422 | 87.2 | 416 | 86.0 | 1 | |
| Yes | 62 | 12.8 | 68 | 14.1 | 0.89 | 0.61 to 1.31 |
| **Myocardial infarction** | | | | | | |
| No | 429 | 88.6 | 436 | 90.1 | 1 | |
| Yes | 55 | 11.4 | 48 | 9.9 | 1.16 | 0.78 to 1.72 |
| **Congestive heart failure** | | | | | | |
| No | 459 | 94.8 | 451 | 93.2 | 1 | |
| Yes | 25 | 5.2 | 33 | 6.8 | 0.72 | 0.41 to 1.27 |
| **Peripheral vascular disease** | | | | | | |
| No | 452 | 93.4 | 462 | 95.5 | 1 | |
| Yes | 32 | 6.6 | 22 | 4.6 | 1.53 | 0.86 to 2.72 |
| **Cerebrovascular disease** | | | | | | |
| No | 433 | 89.5 | 440 | 90.9 | 1 | |
| Yes | 51 | 10.5 | 44 | 9.1 | 1.18 | 0.77 to 1.81 |
| **COPD** | | | | | | |
| No | 402 | 83.1 | 397 | 82.0 | 1 | |
| Yes | 82 | 16.9 | 87 | 18.0 | 0.98 | 0.64 to 1.50 |
| **Connective tissue disorder** | | | | | | |
| No | 441 | 91.1 | 440 | 90.9 | 1 | |
| Yes | 43 | 8.9 | 44 | 9.1 | 0.98 | 0.64 to 1.50 |
| **Diabetes without complications** | | | | | | |
| No | 420 | 86.8 | 429 | 88.6 | 1 | |
| Yes | 64 | 13.2 | 55 | 11.4 | 1.20 | 0.81 to 1.77 |
| **Peptic ulcer** | | | | | | |

**Table 2** Continued

| Characteristics | Case | | Control | | OR* | 95% CI |
|---|---|---|---|---|---|---|
| | n | % | n | % | | |
| No | 446 | 92.2 | 456 | 94.2 | 1 | |
| Yes | 38 | 7.9 | 28 | 5.8 | 1.40 | 0.84 to 2.34 |
| Liver disease | | | | | | |
| No | 481 | 99.4 | 481 | 99.4 | 1 | |
| Yes | 3 | 0.6 | 3 | 0.6 | 1 | 0.20 to 4.95 |
| Hemiplegia/Paraplegia | | | | | | |
| No | 484 | 100.0 | 482 | 99.6 | 1 | |
| Yes | 0 | 0.0 | 2 | 0.4 | – | – |
| Renal disease | | | | | | |
| No | 441 | 91.1 | 449 | 92.8 | 1 | |
| Yes | 43 | 8.9 | 35 | 7.2 | 1.24 | 0.79 to 1.94 |
| Diabetes with complications | | | | | | |
| No | 465 | 96.1 | 460 | 95.1 | 1 | |
| Yes | 19 | 3.9 | 24 | 5.0 | 0.79 | 0.43 to 1.45 |
| Cancer | | | | | | |
| No | 468 | 96.7 | 456 | 94.2 | 1 | |
| Yes | 16 | 3.3 | 28 | 5.8 | 0.57 | 0.31 to 1.06 |
| Leukaemia | | | | | | |
| No | 482 | 99.6 | 482 | 99.6 | 1 | |
| Yes | 2 | 0.4 | 2 | 0.4 | 1.00 | 0.14 to 7.10 |
| Lymphoma | | | | | | |
| No | 482 | 99.6 | 483 | 99.8 | 1 | |
| Yes | 2 | 0.4 | 1 | 0.2 | 2.00 | 0.18 to 22.06 |
| Severe liver disease | | | | | | |
| No | 484 | 100 | 483 | 99.8 | 1 | |
| Yes | 0 | 0 | 1 | 0.2 | – | – |
| Metastatic cancer | | | | | | |
| No | 451 | 93.2 | 436 | 90.0 | 1 | |
| Yes | 33 | 6.8 | 48 | 9.9 | 0.64 | 0.40 to 1.04 |

*Conditional logistic regression ORs for cases and controls matched on age, sex and year diagnosis.

(table 3). Compared with Dukes stage A, Dukes stage D and unknown stage were associated with early death. Unknown histological type, unspecified anatomical site and undetermined grade were also associated with early death. Patients with a family history of colorectal cancer had lower odds of early death compared with patients without a family history (table 4).

## Multivariable analysis
Unknown marital status, being single, widowed, divorced and living alone were all associated with early death compared with patients who were married/cohabiting after adjusting for other patient characteristics and comorbidities. Regarding socioeconomic status, a deprivation gradient for early death was apparent in older people living within quintile 5. This relationship existed for all patients in quintile 4. These socially deprived groups had higher odds of early death compared with the least deprived after adjusting for other factors (table 5). Being underweight between 1 and 3 years before diagnosis was significantly associated with early death in multivariable analysis. Unspecified histology and Dukes stage D disease remained positively associated with early death in multivariable analysis. Additional models that adjusted for pathway characteristics (attendance at A&E and number of GP consultations prior to diagnosis) did not explain the association between marital status and early death (p<0.01). Influenza vaccination attendance and baseline consultation activity, dementia, psychiatric illness and smoking status were not significantly associated with early death in multivariable analysis.

**Table 3** Psychiatric illness among cases and controls and associated clogit ORs and 95% CIs

| Characteristics | Case | | Control | | OR* | 95% CI |
|---|---|---|---|---|---|---|
| | n | % | n | % | | |
| Learning disability | | | | | | |
| No | 479 | 99.0 | 482 | 99.6 | 1 | |
| Yes | 5 | 1.0 | 2 | 0.4 | 1.24 | 0.85 to 1.79 |
| Anxiety/Depression | | | | | | |
| No | 419 | 86.6 | 425 | 87.8 | 1 | |
| Yes | 65 | 13.4 | 59 | 12.2 | 1.13 | 0.76 to 1.70 |
| Schizophrenia | | | | | | |
| No | 480 | 99.2 | 479 | 99.0 | 1 | |
| Yes | 4 | 0.8 | 5 | 1.0 | 0.80 | 0.22 to 2.98 |
| Other psychiatric disorder | | | | | | |
| No | 470 | 97.1 | 480 | 99.2 | 1 | |
| Yes | 14 | 2.9 | 4 | 0.8 | 3.50 | 1.15 to 10.63 |

*Conditional logistic regression ORs for cases and controls matched on age, sex and year diagnosis.

Consistent patterns were observed among men and women when data were stratified by sex, although the association between deprivation and early death was strongly pronounced among women and not present among men. Age stratification showed the odds of early death were higher for those living in rural areas compared with urban areas among patients aged ≥78 years. This association was not apparent in those aged <78. Otherwise, there were no significant differences in the factors associated with early death between those aged <78 and those aged ≥78 (see table 5—only data for ≥78 shown).

## DISCUSSION
### Summary of findings
This study investigated the characteristics of patients who died within 3 months of a diagnosis of colon cancer, by way of univariate and multivariate analyses, with the aim of identifying specific patient factors that can be addressed or accounted for to improve survival outcomes. Social isolation was identified as a common characteristic of early death in patients with colon cancer. The different forms of social isolation studied included living alone and being unmarried (as opposed to cohabiting with a partner), residing in more deprived communities (as opposed to living in quintile 1–2 communities), living in a rural area when elderly (as opposed to an urban area), and having dementia or a psychiatric illness. Each of these factors was comparatively associated with early death. Previous studies suggested that poorer outcomes for unmarried people and those living alone were mediated through weak social support,[27] which exerted its influence on outcomes through later presentation.[28] This view is consistent with several other studies that reported more negative cancer beliefs,[29] lower symptom awareness and greater perceived barriers to GP help-seeking among this group.[30] While weaker social support has previously been associated with later stage disease,[31 32] in this study, despite the collection of a range of detailed pathway and treatment variables, the association between marital status, accommodation status and early death remained unexplained. Other studies suggested biopsychosocial explanations for poorer cancer outcomes in unmarried patients with cancer, including chronic stress[33] and weaker immune response.[34] Marital status and accommodation status have an important association with cancer and health outcomes generally,[35] yet it is a research area that remains relatively underinvestigated.

As in other studies, deprivation quintile was associated with early death, with a gradient in the odds of early death with increasing deprivation score.[4 36 37] However, unlike other studies, this association seems restricted to women. Like marital/accommodation status, there was little evidence from this study suggesting that the association between deprivation and early death is explained by characteristics of the pathway to diagnosis as the associations persisted after adjusting for attendance at A&E in the pathway to diagnosis, Dukes stage and GP episodes in the 3 months to diagnosis. There is no direct biological basis for an association between deprivation and survival. Mediating factors may include lower performance status due to higher tobacco consumption,[38] lower uptake of treatments due to fatalistic cancer beliefs[39] or differential access to services.[40] The association between rural residence and early death among the oldest patients was independent of deprivation and strongly significant. This is likely to relate to either isolation, access to services or both. This is a well-defined target group for early diagnosis interventions, but further study is required to investigate the link between lack of social contact and cancer survival if one exists.

**Table 4** Disease characteristics among cases and controls and associated ORs and 95% CIs

| Characteristics | Case n | % | Control n | % | OR* | 95% CI |
|---|---|---|---|---|---|---|
| **Anatomical location** | | | | | | |
| Ascending | 52 | 10.7 | 57 | 11.8 | 1 | |
| Caecum | 93 | 19.2 | 136 | 28.1 | 0.83 | 0.52 to 1.29 |
| Other | 90 | 18.6 | 93 | 19.2 | 1.19 | 0.73 to 1.93 |
| Descending | 29 | 6.0 | 18 | 3.7 | 1.97 | 0.97 to 3.99 |
| Sigmoid colon | 95 | 19.6 | 132 | 27.3 | 0.86 | 0.54 to 1.37 |
| Not specified | 125 | 25.8 | 48 | 9.9 | 2.96 | 1.75 to 5.02 |
| **Histological type** | | | | | | |
| Adenocarcinoma | 256 | 52.9 | 384 | 79.3 | 1 | |
| Mucinous | 14 | 2.9 | 28 | 5.8 | 0.91 | 0.45 to 1.81 |
| Not specified | 198 | 40.9 | 60 | 12.4 | 5.62 | 3.80 to 8.34 |
| Other | 16 | 3.3 | 12 | 2.5 | 1.68 | 0.77 to 3.67 |
| **Metastases** | | | | | | |
| None | 37 | 7.6 | 64 | 13.2 | 1 | |
| Bone | 9 | 1.9 | 1 | 0.2 | 17.2 | 2.48 to 143.6 |
| Liver | 134 | 27.7 | 108 | 22.3 | 2.37 | 1.42 to 3.95 |
| Lung | 14 | 2.9 | 10 | 2.1 | 2.55 | 1.01 to 6.38 |
| Other | 27 | 5.6 | 23 | 4.8 | 2.17 | 1.07 to 4.41 |
| Unknown | 263 | 54.3 | 278 | 57.4 | 1.77 | 1.10 to 2.84 |
| **Dukes stage** | | | | | | |
| A | 8 | 1.7 | 19 | 3.9 | 1 | |
| B | 46 | 9.5 | 85 | 17.6 | 1.37 | 0.54 to 3.48 |
| C | 47 | 9.7 | 127 | 26.2 | 1.02 | 0.41 to 2.52 |
| D | 160 | 33.1 | 140 | 28.9 | 2.96 | 1.23 to 7.14 |
| Unknown | 233 | 46.1 | 113 | 23.4 | 5.65 | 2.30 to 13.91 |
| **Grade (differentiation)** | | | | | | |
| Well/Moderate | 93 | 19.2 | 190 | 39.3 | 1 | |
| Poor/Undifferentiated | 40 | 8.3 | 57 | 11.8 | 1.45 | 0.89 to 2.36 |
| Not determined | 351 | 72.5 | 237 | 50.0 | 3.32 | 2.38 to 4.62 |
| **Colorectal polyps** | | | | | | |
| No | 467 | 10.7 | 57 | 11.8 | 1 | |
| Yes | 17 | 19.2 | 136 | 28.1 | 0.80 | 0.42 to 1.54 |
| **Bowel cancer family history** | | | | | | |
| No | 459 | 25.8 | 48 | 9.9 | 1 | |
| Yes | 25 | 6.0 | 18 | 3.7 | 0.52 | 0.31 to 0.88 |

*Conditional logistic regression ORs for cases and controls matched on age, sex and year diagnosis.

Similar to previous work on early colorectal cancer death,[4] a consistent feature associated with early death was that of incomplete data due to limited diagnostic testing. Missing histology, stage, grade and anatomical site may be explained by very ill patients not receiving complete investigation. These characteristics may therefore be viewed as confounding by indication; as opposed to explaining early death, the associations are explained by early death. The relationships between unknown marital status, accommodation and outcomes are more difficult to explain. Despite the fact that GP records and secondary care databases were searched, and the information relates to a period over a year prior to diagnosis of cancer, a strong association with early death was observed indicating the data were not missing at random. Missing data on living status, smoking status and alcohol consumption were also associated with early death in univariate analysis. Further work is required to explain

**Table 5** Multivariable analysis of the association between patient characteristics and early death

| | | All patients | | | Male | | | Female | | | Aged ≥78 years | | |
|---|---|---|---|---|---|---|---|---|---|---|---|---|---|
| | | n* | OR | 95% CI | n* | OR | 95% CI | n* | OR | 95% CI | n* | OR | 95% CI |
| Marital/Accommodation status | Married and cohabiting | 371 | 1.00 | | 248 | 1.00 | | 123 | 1.00 | | 125 | 1.00 | |
| | Married not cohabiting | 26 | 1.90 | 0.65 to 5.57 | 11 | 1.27 | 0.27 to 6.01 | 15 | 3.72 | 0.70 to 19.7 | 17 | 1.60 | 0.45 to 5.67 |
| | Institution care | 63 | 1.93 | 0.93 to 4.00 | 25 | 1.78 | 0.57 to 5.57 | 38 | 2.60 | 0.85 to 7.82 | 52 | 3.02 | 1.22 to 7.49 |
| | Unknown | 127 | 5.01 | 2.82 to 8.89 | 66 | 3.87 | 1.76 to 8.38 | 61 | 8.49 | 3.12 to 23.1 | 68 | 4.32 | 1.97 to 9.47 |
| | Living alone and SWD | 270 | 1.64 | 1.07 to 2.50 | 113 | 1.55 | 0.86 to 2.80 | 157 | 2.21 | 1.01 to 4.80 | 168 | 1.62 | 0.90 to 2.91 |
| | Other and SWD | 111 | 1.39 | 0.80 to 2.44 | 47 | 1.28 | 0.59 to 2.74 | 64 | 2.03 | 0.76 to 5.44 | 61 | 0.85 | 0.37 to 1.95 |
| Deprivation quintile | Q1 (least deprived) | 153 | 1.00 | | 87 | 1.00 | | 66 | 1.00 | | 83 | 1.00 | |
| | Q2 | 187 | 1.28 | 0.74 to 2.25 | 91 | 0.84 | 0.39 to 1.82 | 96 | 2.90 | 1.10 to 7.6 | 96 | 2.29 | 1.06 to 4.93 |
| | Q3 | 217 | 1.42 | 0.83 to 2.42 | 112 | 1.28 | 0.62 to 2.63 | 105 | 2.09 | 0.81 to 5.4 | 107 | 2.05 | 0.96 to 4.38 |
| | Q4 | 212 | 1.88 | 1.09 to 3.23 | 112 | 1.60 | 0.79 to 3.22 | 100 | 3.10 | 1.14 to 8.4 | 113 | 2.02 | 0.98 to 4.15 |
| | Q5 (most deprived) | 199 | 1.72 | 0.98 to 3.02 | 108 | 1.13 | 0.53 to 2.33 | 91 | 4.70 | 1.68 to 13.2 | 92 | 2.64 | 1.15 to 6.06 |
| Being underweight (BMI <18.5) | | 22 | 3.99 | 1.14 to 14.0 | 6 | 2.74 | 0.32 to 21.1 | 16 | 3.96 | 0.65 to 24.2 | 13 | 7.27 | 1.42 to 68.8 |
| Dukes stage | A | 8 | 1.00 | | 20 | 1.00 | | 7 | 1.00 | | 10 | 1.00 | |
| | B | 46 | 1.85 | 0.64 to 5.32 | 70 | 1.72 | 0.52 to 5.67 | 61 | 4.90 | 0.32 to 75.1 | 69 | 2.43 | 0.43 to 13.7 |
| | C | 47 | 1.51 | 0.54 to 4.27 | 82 | 1.13 | 0.33 to 3.82 | 92 | 6.04 | 0.43 to 83.9 | 88 | 2.17 | 0.41 to 11.6 |
| | D | 160 | 3.07 | 1.13 to 8.38 | 162 | 1.98 | 0.63 to 6.21 | 138 | 13.8 | 1.02 to 188 | 109 | 4.64 | 0.84 to 25.7 |
| | Unknown | 233 | 2.99 | 1.05 to 8.53 | 176 | 1.86 | 0.58 to 6.02 | 160 | 15.8 | 1.06 to 236 | 215 | 4.27 | 0.79 to 22.9 |
| Histology type | Adenocarcinoma | 440 | 1.00 | | 352 | 1.00 | | 288 | 1.00 | | 296 | 1.00 | |
| | Mucinous | 42 | 1.20 | 0.54 to 2.66 | 17 | 2.13 | 0.59 to 7.64 | 25 | 0.87 | 0.27 to 2.72 | 22 | 1.06 | 0.35 to 3.27 |
| | Not specified | 258 | 3.66 | 2.17 to 6.17 | 126 | 5.02 | 2.47 to 10.2 | 132 | 2.01 | 0.82 to 4.90 | 166 | 2.70 | 1.37 to 5.31 |
| | Other | 28 | 1.16 | 0.48 to 2.83 | 15 | 1.58 | 0.44 to 5.66 | 13 | 0.76 | 0.17 to 3.46 | 7 | 0.58 | 0.09 to 3.77 |
| Rural residence | | 344 | 1.40 | 0.98 to 2.01 | 182 | 1.29 | 0.78 to 2.17 | 162 | 1.53 | 0.86 to 2.74 | 178 | 1.96 | 1.21 to 3.17 |

*n, cases and controls combined.
BMI, body mass index; SWD, single, widowed, divorced.

this relationship, with possible areas for investigation including the patient–practitioner relationship.

While comorbidity was a common feature of patients in this study, we did not find it associated with dying within 3 months of diagnosis. Previous studies have observed that comorbidity exerted the greatest influence in the later phases of the survival pathway,[41] although CCI score has been observed as an independent indicator of early death elsewhere.[4] We had matched cases and controls on age, and as comorbidity is strongly linked to age this may have reduced our ability to detect this as an independent factor. In addition, being underweight had a strong independent association with early death. This was measured between 1 and 3 years before diagnosis, and it is likely either related to disease progressing over a longer time period or to poorer performance status. However, as only 17 cases were described as underweight, it explains less than 4% of the total early deaths.

While the CCI was not associated with early death, dementia and other psychiatric illness, as individual comorbid conditions, were associated with early death. However, in multivariable analysis, these relationships did not persist. Those with dementia were more likely to have missing data on stage and anatomical location, perhaps suggesting that these patients were less likely to be investigated for their disease. Similar findings have been reported in other colon cancer studies, with dementia associated with poorer colorectal cancer outcomes, later stage disease[42] and less invasive investigation.[43] The relationship between other psychiatric illness and early death was also attenuated by stage, again suggesting a role of diagnostic intervals on the pathway to diagnosis explaining early death. Underlying causes of delay may relate to symptom recognition by carer, patient, practitioner, as well as patient communication or competing healthcare priorities. A key objective of this study was to determine if patient health-seeking characteristics were associated with early death following colon cancer diagnosis. Uptake of the influenza vaccine, baseline consultation activity and non-attendance at appointments were identified as three easily captured indicators of health-seeking behaviour. It was hypothesised that patients with more regular or more compliant health-seeking behaviour would have better outcomes than those without, mediated through longer diagnostic intervals and later stage disease.

With regard to health-seeking behaviour, although attendance for the influenza vaccine was inversely associated with early death, this association was not significant in multivariable analysis. While attendance for the vaccine may be considered an indicator of health compliance, it is likely to reflect various health attitudes and behaviours, with previous UK studies reporting several social and cultural factors that were associated with influenza vaccination uptake, as well as perception regarding health status and susceptibility to influenza.[44] The other indicator of health-seeking, frequent GP attendance, was correlated with comorbidity in this study, a relationship reported in several other research studies.[10 45]

Although ex-smokers were identified in univariate analysis as having non-significantly lower odds of early death, this association was largely explained in the minimally adjusted model with little evidence to suggest that poorer outcome among smokers is real. While previous studies have demonstrated a significant association between alcohol-related hospital admissions and early death following a colorectal cancer diagnosis,[37] this association was not observed in the current study possibly because this study was unable to discriminate between heavy and moderate alcohol use.

### Strengths and weaknesses

The study sample was generated from a high-quality population-based cancer registry system with relatively few DCO cases[46] with full access to general practice records and hospital clinical records. While missing data were a feature of the study, rather than acting as an impediment to our understanding of characteristics of early death, this appears to be one of the defining features of this patient group. Another recent study of early death has reported a similar pattern of high levels of missing data in those with the poorest outcomes.[4] However, the fact remains that missing data were a feature of this study, and potential improvements to the way data are recorded in the NICR are continuously being made. The study investigated a broad range of factors that may be associated with early death and allowed for adjustment of a range of confounding factors such as comorbidity, smoking and alcohol status. We present results from both univariate and multivariate analyses but place greater emphasis on the results from the multivariate analysis due to the complex nature of interacting factors causing early death from colon cancer. The recording of the patient characteristics that vary over time at between 1 and 3 years before diagnosis was an important feature of the study design. Cancer diagnosis has been previously identified as being associated with changes in health-seeking behaviour, comorbidity, BMI and lifestyle factors. Recording these factors based on over 1 year before diagnosis strengthens the assessment of causal inference between variables and early death.

The case–control design of the study provided an efficient methodology to collect data and allowed the development of a control group that was matched on important but non-modifiable characteristics. While age, sex and year of diagnosis were fixed in the current study, their interaction with diagnostic pathway features and other characteristics could not be investigated. Previous studies have shown longer diagnostic timelines, later stage disease at diagnosis, lower symptom awareness and more negative cancer beliefs to be variable depending on age[47] and sex.[48] In addition, the matching on age may have reduced variation in other characteristics such as comorbidity. The study population was also selected in a period before the introduction of screening; therefore, all patients in both the case and control groups were clinically diagnosed. While bowel screening was introduced

in NI in 2011[22] and now represents an important pathway in cancer diagnosis, the majority of patients are still diagnosed clinically and the early clinical detection of symptoms remains both important and relevant—particularly for cancer stage and, by extension, cancer survival. Future iterations of this work would likely benefit from matching of cases and controls on cancer stage, in addition to age, sex and year of diagnosis.

The use of a control group of longer survivors provided useful comparative information to investigate risk of early death, with the choice of a control sample of deceased patients with colon cancer removing any risk of consent bias from the study. However, while a buffer of 3 months was placed between cases and controls to allow better discrimination between the two, the survival of controls was restricted to a population of patients whose survival was less than 3 years and similar to the case population. Despite this, the study was able to identify several characteristics which discriminated between cases and controls suggesting that the patients who die within the first few months of diagnosis are a specific patient cohort who requires attention.

## CONCLUSIONS

This comprehensive study of early death from colon cancer has identified several population subgroups that warrant special attention. These include those who are single, living alone, older people living in rural environments, and people from the most deprived communities as well as those living in residential or nursing care. These likely comprise some of the most isolated people in society. However, while the aforementioned variables are an indicator of social isolation, this study was not designed to actually *investigate* isolation (ie, lack of social contact or poor social support networks). Therefore, further study is required to confirm that social isolation is definitely linked with poor cancer survival outcomes. Further studies are also required to better understand the role of missing data in patient records. Furthermore, additional work ought to be undertaken to determine if these patterns are consistent in other International Cancer Benchmarking Partnership (ICBP) countries. Finally, because of increased colon cancer survival, future studies investigating risk factors for an *early death* using a case–control methodology would likely benefit from comparing cases who suffer early mortality with controls who survive beyond 5 or perhaps even 10 years as opposed to the 3-year survival control group used in this study.[49–51]

**Acknowledgements** The authors gratefully appreciate the contribution of data abstractors: Donna Floyd, Rosemary Ward, Jacqui Napier, Kate Donnelly and Bríd Morris. The authors also thank Business Services Organisation, Health and Social Care NI for facilitating the note review. Finally, the authors would like to thank all patients whose data was used in this study.

**Contributors** CD designed the study, undertook all statistical analyses and drafted the manuscript. NH was the medical adviser for the study, provided advice on statistical analysis and interpretation, and contributed to the writing of the manuscript. ADM contributed to the writing, editing and submission of the paper. MD provided advice and guidance on the study design and contributed to the writing of the manuscript. LA provided advice and guidance on the study design and contributed to the writing of the manuscript. LR contributed to the study design, development of data collection forms and data abstractor training, and analysis and interpretation of the results. AG designed the study, was the principal investigator, funding recipient and contributed to the writing of the manuscript.

**Funding** The TEDI study was funded by the Cancer Research UK led National Awareness & Early Diagnosis Initiative (Principal Investigator: AG, Co-Investigators: CD, NH, ADM, MD, LR LA). The Northern Ireland Cancer Registry is funded by the Public Health Agency Northern Ireland (Principal Investigator: AG)

**Competing interests** None declared.

**Patient consent for publication** Not required.

**Ethics approval** Ethical approval for this study was granted by the Office of Research Ethics Committees Northern Ireland (12/NI/0034). This committee receives input from lay member(s) of the public before reaching a decision about whether or not to approve research studies.

**Provenance and peer review** Not commissioned; externally peer reviewed.

**Data sharing statement** Data stored within the archives of the N Ireland Cancer Registry and available with permission to researchers on secondment to the registry.

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
