## [Reviewer comments · BMJ Open]

ARTICLE DETAILS

TITLE (PROVISIONAL)	Predictors of an early death in patients diagnosed with colon cancer: a retrospective case-control study in the United Kingdom
AUTHORS	Donnelly, Conan; Hart, Nigel; McCrorie, Alan; Donnelly, Michael; Anderson, Lesley; Ranaghan, Lisa; Gavin, Anna

VERSION 1 - REVIEW

REVIEWER	Natalia Calanzani University of Edinburgh, UK
REVIEW RETURNED	30-Sep-2017

GENERAL COMMENTS	This is an interesting manuscript in the field of early diagnosis, where evidence is still much needed. The paper is well written, with very clear language. I am not a statistician, but to my knowledge the chosen analysis is adequate for what the authors proposed to do. Nonetheless, I believe that the manuscript could benefit from a few clarifications (please see below). 1. I believe that the main issue to be addressed is the assumption that patients living alone, in rural areas and in deprived areas are necessarily in social isolation (i.e. lack of social contact, support or interactions with the community). Although it is quite reasonable to expect that patients in social isolation are more likely to live alone or in remote areas, it does not mean that people living alone, in rural areas or in deprived regions are necessarily socially isolated (unless there is evidence that this is the case for the population in this study). There was no specific assessment of whether the patients lacked social contact, for example. I would consider changing the way the findings, discussion and manuscript title are described regarding this issue. Perhaps the authors could make it clearer that these variables may be an indication of social isolation, but since the latter was not specifically investigated, other studies are needed to confirm if this is the case. The authors have already done so in the first paragraph of page 16 (hypothesising a relationship between rural residence and social isolation), but elsewhere (including the title of the paper) a different approach is taken.2. Methodology, page 5, rows 18-20 and 35-36: could the authors specify which principles from the Aarhus statement were used? Furthermore, does the NICR also follow Aarhus guidance for date of diagnosis or do they take a different approach?3. Page 7, first paragraph: consider adding the interquartile range (IQR) for age (median age 78 is mentioned, but no IQR is given). Supplementary table S1 only gives the mean and the standard
--

	deviation. Information on the minimal and maximum age of included patients would also be useful. 5. Page 8, Table 1 and Page 14, Table 5: Were marital/housing status (also described in text as accommodation status) merged in the multivariable analysis because of multicollinearity (Table 5)? Consider adding to the methods why recoding was needed 6. Page 11, row 2, and other sections in the manuscript approaching missing data: consider adding something to the discussion about the need to improve data recording. Furthermore, consider making it clearer that missing data was a study limitation, even though the issue generated interesting findings. At the moment the discussion (subsection Strengths and weaknesses) reads as if missing data was only beneficial to the study. 7. Discussion, page 15, first paragraph: The discussion begins with a sentence stating that the study “conducted a comprehensive examination of the pathway to diagnosis for colon cancer patients”. This is somewhat different from the aim described elsewhere (abstract and last paragraph in the Introduction). The study did not discuss some important issues for pathways to diagnosis (such as source of referral). Consider rephrasing this sentence according to the study aims. Some minor issues the authors may wish to check:  - Several tables: It seems that significant variables are not consistently greyed out in the tables - Page 14, Table 5: There seems to be a typo after the variable Underweight (+ve). If this is correct, +ve may need to be explained as a footnote. - Page 9, sentence starting with “Being underweight” – it seems that something happened to this sentence; it is a bit confusing as it is - Page 15, discussion, second paragraph, third sentence: this is slightly long and it is a bit confusing - Page 19, second paragraph, last work – there is a typo here (i.e. relavent instead of relevant)
--	--

REVIEWER	Turid Follestad Department of Public Health and Nursing, Faculty of Medicine and Health Sciences, Norwegian University of Science and Technology Norway
REVIEW RETURNED	03-Nov-2017

GENERAL COMMENTS	This review is made from a statistician’s point of view, with particular emphasis on the statistical methods and analyses used. The study was conducted using an individually matched case-control design, and aimed at identifying factors associated with early death from colon cancer. A major strength of the study is the high-quality data, and relatively large sample size. The selected method for the statistical analysis, conditional logistic regression (CLR), is appropriate for the study design, due to the individual matching of cases and controls. The subgroup analyses for age and sex are also sound. However, there are some issues regarding the presentation and interpretation of the results that should be given consideration. In addition, the subgroups could have been
--

compared directly by an extension of the CLR model, representing an improvement over subgroup analysis, for which no such direct comparisons can be made. The issues are detailed below, organized by sections in the manuscript.

Title:

1. Does «older» refer to a characteristic of the study population, or to older age being associated with early death? The results do not support the latter interpretation; see further comments below.

Abstract and Key message:

2. The listed numerical values for the ORs with 95% CIs do not match the values given in Table 5.

3. p.2, l. 24: “being underweight compared to normal weight”.

According to Tables 2 and 5, the reference category seems to be “normal or obese”, and not “normal”. Similarly, the reference category for “obese” seems to be “normal or underweight”.

4. p.2, l. 26: From the subgroup analysis it can be concluded that rural, as compared to urban, residence is a predictor among the older patient group. It is not clear whether this is what the authors mean by “older patients living in rural areas were independent predictors”.

5. Key message: According to Table 5 «rural residence» should apply to the older age group only.

Methodology:

6. p.6, l. 40-44. The authors should state clearly how the sample size calculation, implying marital status as a main factor of interest, relates to the main aim of the study as given on p.5, line 1-8. In addition, the statistical method for which the calculation is performed ought to be mentioned.

7. p.6, l. 53-57: This sentence could mistakenly be interpreted to mean that univariate, unadjusted analyses could be used to identify independent factors associated with early death. A restructuring of the whole paragraph would help clarify that independent factors were identified from the multivariate analysis.

8. The selected level of significance should be stated. In addition, the authors should argue why adjustment for multiple testing was not considered. Even though formal adjustment is not included, the authors ought to address this issue in the Methodology section, and when interpreting the results.

Results:

9. The authors have, in accordance with the STROBE checklist, presented “unadjusted” as well as “confounder-adjusted” OR estimates. However, since the analysis aimed at identifying independent predictors of early death, found based on the multivariate analysis, the results section would benefit from putting less emphasis on the results from the univariate (Tables 1-4) and more on those from the multivariate (Table 5) analysis. This comment also applies to the discussion section.

10. Candidate variables to be included in the multivariate analysis ought not to be selected based on results from univariate analyses alone, but also based on clinical knowledge of relevant variables.

11. p.8, l. 1-2. The expression “approached statistical significance” should not be used. Given the significance level, results are either significant or not.

12. For clarification, p-values could be added in all tables, in particular since p-values were used to guide the selection of

	variables, and since no fromal adjustment for multiple testing was included. 13. The tables present OR referred to given reference categories and no overall p-values are presented. Does that imply that the authors consider pairwise comparisons to the selected reference categories to be the only pairwise comparisons of interest? 14. p.9, l.4. Would it be more interesting to comment on the non-significant OR for smokers vs non-smokers rather than focussing on the results for “unknown”? 15. p.11, l. 10. “Other psychiatric conditions”: The authors ought to consider the small counts and the wide CI for this category in their interpretation. 16. p.12, l. 49: “the two most deprived quintiles had significantly higher odds ...”: This applies only to Q4, not Q5 (All patients, Table 5). Discussion: 17. Even though the authors are in general careful not to interpret different results in subgroups (Table 5) as significant differences between the subgroups, the interpretation of subgroup results could be clarified at places, e.g. at p.15, l.10 where it is not clear what is the reference. 18. p.18, l. 20-22: From Table 1 it does not seem correct to conclude that the odds for “ex-smokers” is significantly different from that of “non-smokers” in the univariate analysis. 19. p. 20, l. 22-27. It is not correct that interaction effects involving age, sex and year of diagnosis (i.e. matching variables) cannot be estimated in the CLR model, even though the main effects of the matching variables cannot. Including interactions in the CLR model, the model will provide estimates and CIs for the ratio of ORs between age groups, and between males and females. Since these estimates cannot be provided, at least not directly, by the subgroups analyses, the CLR model with interactions would be beneficial. Conclusion: 20. p. 20, l. 14-15: «who are older,»: The results in Table 5 does not support the conclusion that early death is associated with older age. 21. p. 20, l. 18-19: “.. and those with demenita and psychiatric disorders.” The authors have not given evidence that these are independent predictors. A significant association with early death was found in univariate analyses only (Tables 2 and 3), and regarding psychiatric disorders for “Other psychiatric disorders” only. MINOR ISSUES: 1. p.6, l.43: Does “10% difference” more precisely mean a 10 percent points difference? 2. p. 9, l. 11-12: There are some misprints in this sentence. 3. p.9, l.18: “(p=0.32)”. Add which statistical test that has been used here. 4. A “Statistical method” heading should be included in the Methodology section. 5. Table 4: % is missing for Grade: Not determined.
--	---

VERSION 1 – AUTHOR RESPONSE

Reviewer 1 – Natalia Calanzani

Thank you for your comments and suggestions. The main issue you wished us to address was our assumption that patients living alone, in rural areas, or in socially deprived circumstances were 'isolated'. Our original manuscript including title, findings, and discussion were structured based upon this assumption. We have modified all of the above to place less emphasis upon social isolation alone and more upon the multiple factors associated with early death. We have also suggested (both in our abstract and main body of paper) that further investigation into social isolation is necessary before coming to any conclusions about causation. Your other comments included a suggestion to list which principles of the Aarhus statement were adhered to, inclusion of IQR for our sample population, and modification of discussion to include a statement about missing data have all been taken into account and manuscript modified accordingly.

Reviewer 2 – Turid Follestad

Thank you for your comments and suggestions. The main issues you wished us to address were our presentation of statistical results including within our abstract, tables, and main body of text. First and foremost, we have modified the manuscript title to better suit study results and conclusions (i.e. less emphasis placed upon the 'social isolated older patient's' aspect of the title and abstract and more emphasis placed upon the multifactorial causes of early death in colon cancer as evidenced by both univariate and multivariate analyses. Odd's ratios within our abstract now match those found within table 5 and reference categories within table 2 have been amended to underweight or obese. You suggested that we place less emphasis upon our univariate analysis as compared with multivariate analysis. We have taken these comments on board and made amendments to our discussion section but tables 1-4 remain. Other miscellaneous issues addressed include the inclusion of a heading 'statistical methods' within our methodology, correction of 10% difference to 10 percentage point difference, a comment regarding choice of variables of interest included within multivariable analysis being of clinical and statistical significance. Finally, we have modified our conclusion to better reflect results presented within table 5 (i.e. that older patients living in rural residency as opposed to simply older patients die early from colon cancer). Other minor typographical errors you highlighted have been amended also.

VERSION 2 – REVIEW

REVIEWER	Dr Linda Williams University of Edinburgh, UK
REVIEW RETURNED	20-Nov-2018

GENERAL COMMENTS	An interesting, well-written paper. However, I have a few concerns. 1) The data are now quite old, even with accommodating the three year survival for the controls. Is this deliberate to avoid the introduction of screening? 2) I was puzzled over the choices for the sample size calculation, these are not explained well. You then appeared to immediately ignore this calculation, by taking the 600 first and then removing the various exclusions. If 600 evaluable patients were required,
---

	why did you only analyse 484? Additionally, given your complete eligible population was only 699, why not analyse all of them? You might have been able to reach closer to you target sample size that way. 3) Table 2. it is unclear what your comparator group is in the BMI analysis. You state in the text that underweight BMI is compared to normal BMI, but the numbers in the table do not bear that out. 4) Table 5 is unreadable 5) As stage is such a strong predictor of death, it might have been interesting to have matched the cases and controls on stage as well.
--	---

REVIEWER	Harm Rutten Catharina Hospital Eindhoven, and, GROW, school of oncology and developmental biology, University Maastricht, Maastricht, The Netherlands
REVIEW RETURNED	28-Jan-2019

GENERAL COMMENTS	The paper is well written and is good to understand. It is an important question. My major concern is the chosen method. The authors compare early mortality to less early mortality, but still mortality. In my opinion a patient operated with curative intent should not die of early mortality, but again also not from a little bit later mortality. I.e. mortality after 6 months is still quite early mortality and should not occur. Why is mortality between surgical procedure and 6 months not compared to a group who did not suffer from mortality? This would be easy to understand. If the authors would like to compare early and less early postop mortality, I would suggest to add an extra group with no mortality
---

VERSION 2 – AUTHOR RESPONSE

Reviewer 1 – Linda Williams, University of Edinburgh

1) Yes, you are correct. An explanatory sentence has been added on page 3 of the revised manuscript document (bullet point 3) explaining that our analysis mitigates the effect of the introduction of bowel cancer screening. This point is mentioned once again in the 2nd paragraph of our methodology section.

2) A more understandable sample size rationale is now included in a similar format to other casecontrol studies published within BMJ Open. Final paragraph, page 6, justifies our choice of sample size included within this study using terminology we hope readers can understand. Recruitment of 960 participants (480 cases and 480 controls) allows this study to detect an OR between 2.1 and 1.5 depending on prevalence of risk factor under investigation.

3) Comparator group in BMI analysis is non-underweight BMI (i.e. a combination of normal weight, overweight, and obese BMI individuals). Table 2 modified including numerical totals to better reflect this fact.

4) Table 5 completely reworked to be more readable.

5) We agree. A sentence has been added to page 22 paragraph 2 to reflect this observation and our desire to address this in future iterations of this work.

Reviewer 2 – Harm Rutten, University of Maastricht

Thank you for your comments. We are in agreement with your observation that cases would benefit from being compared to controls who live longer than 6-36 months. Indeed, with recent improvements in survival trends, future iterations of this work done on European/North American patients would probably benefit from cases being compared with controls who survive for 5 years, 10 years, or those still alive. We have added a sentence and 3 additional references to our conclusion paragraph (page 23) to reflect your recommendation. The reason why we did not do this in the original analysis is based upon N. Ireland data, which shows that between 2005-2010, people who were diagnosed with colon cancer had a less than 50% chance of being alive at 5 years post diagnosis. Therefore, survival time of 3 years was selected.

VERSION 3 – REVIEW

REVIEWER	Dr Linda Williams University of Edinburgh, UK
REVIEW RETURNED	01-Apr-2019

GENERAL COMMENTS	I am happy with the improvements made.
--